# Optimal Contact Position of Subthalamic Nucleus Deep Brain Stimulation for Reducing Restless Legs Syndrome in Parkinson’s Disease Patients: One-Year Follow-Up with 33 Patients

**DOI:** 10.3390/brainsci12121645

**Published:** 2022-12-01

**Authors:** Hongbing Lei, Chunhui Yang, Mingyang Zhang, Yiqing Qiu, Jiali Wang, Jinyu Xu, Xiaowu Hu, Xi Wu

**Affiliations:** 1Department of Neurosurgery, the First Affiliated Hospital of Naval Medical University, No. 168 Changhai Road, Yangpu District, Shanghai 200433, China; 2Department of Chemistry, University of Utah, 201 Presidents’ Cir, Salt Lake City, UT 8412, USA

**Keywords:** Parkinson’s disease (PD), restless legs syndrome (RLS), deep brain stimulation of the subthalamic nucleus (STN-DBS), effective stimulation sites

## Abstract

**Objectives:** To determine the short- and medium-term therapeutic effects of subthalamic nucleus (STN) deep brain stimulation (DBS) on restless legs syndrome (RLS) in patients with Parkinson’s disease (PD) and to study the optimal position of activated contacts for RLS symptoms. **Methods:** We preoperatively and postoperatively assessed PD Patients with RLS undergoing STN-DBS. Additionally, we recorded the stimulation parameters that induced RLS or relieved RLS symptoms during a follow-up. Finally, we reconstructed the activated contacts’ position that reduced or induced RLS symptoms. **Results:** 363 PD patients were enrolled. At the 1-year follow-up, we found that the IRLS sum significantly decreased in the RLS group (preoperative 18.758 ± 7.706, postoperative 8.121 ± 7.083, *p* < 0.05). The results of the CGI score, MOS sleep, and RLS QLQ all showed that the STN-DBS improved RLS symptoms after one year. Furthermore, the activated contacts that relieved RLS were mainly located in the central sensorimotor region of the STN. Activated contacts in the inferior sensorimotor part of the STN or in the substantia nigra might have induced RLS symptoms. **Conclusions:** STN-DBS improved RLS in patients with PD in one year, which reduced their sleep disorders and increased their quality of life. Furthermore, the central sensorimotor region part of the STN is the optimal stimulation site.

## 1. Introduction

Parkinson’s disease (PD) is the second most common neurodegenerative disease, whose core motor symptoms are characterized by bradykinesia, akinesia, rigidity, tremor, postural instability, and gait difficulties [1]. PD can also be complicated by a variety of non-motor symptoms such as restless legs syndrome (RLS) [2]. Restless legs syndrome (RLS), also referred to as Willis–Ekbom disease, is a chronic neurological disorder that affects motor activity and the quality of patients’ life [3]. People with RLS have an impulse to move their legs, as well as soreness, cramping, the feeling of ants crawling and other unpleasant sensations in their legs; these symptoms often occur at rest, especially in the evening, and they decrease during motor activity [4]. However, some studies have stated that the RLS risk is higher among PD patients than healthy individuals [5]. The treatment measures for RLS symptoms in PD patients mainly involve dopaminergic agents, opioids, Alpha-2-delta ligands, anticonvulsants, sedative-hypnotics, and deep brain stimulation (DBS) [6].

The efficacy of DBS in relieving motor symptoms and increasing quality of life in PD patients is well established [7]. Deep brain stimulation of the subthalamic nucleus (STN-DBS) can remarkably reduce the symptoms of restless legs syndrome and relieve motor symptoms, and some patients even experience complete symptom relief [8,9,10,11]. However, other studies have found that patients’ RLS symptoms worsened after STN-DBS [12,13].

These controversies may result from the fact that the exact pathophysiological RLS mechanisms are still not fully understood. In addition, the number of related studies and cases on the optimal activation of neural structures and neural pathways by STN-DBS in reducing RLS is relatively small. Therefore, many problems regarding the clinical application of STN-DBS in the treatment of RLS symptoms combined with PD still exist. These include: 1. how to maximize the benefits of STN-DBS for RLS symptoms through comprehensive treatment; 2. the effective stimulation region and stimulation parameters for STN-DBS for relieving RLS; and 3. whether the reduction in RLS symptoms by STN-DBS is durable, and whether there will be RLS symptom fluctuations.

To answer the above questions, we conducted a retrospective observational study to analyze the postoperative reduction in RLS in PD patients who were treated with STN-DBS at the First Affiliated Hospital of Naval Medical University from 1 January 2018 to 31 December 2020. Furthermore, we analyzed the optimal activated contact location of STN-DBS for RLS symptoms that could provide the basis and facilitation for postoperative programming.

## 2. Materials and Methods

### 2.1. Inclusion Criteria

This retrospective clinical study included all the patients who underwent bilateral STN-DBS for PD at the First Affiliated Hospital of Naval Medical University (Shanghai, China) from 1 January 2018 to 31 December 2020. The First Affiliated Hospital of Naval Medical University Ethics Committee (CHEC2018-022) approved this study. The patients who underwent a preoperative evaluation and completed a comprehensive preoperative examination were divided into RLS and non-RLS groups according to the presence or absence of RLS before enrollment. Patients with PD and RLS were enrolled in the RLS group and those without RLS were assigned to the non-RLS group. The inclusion criteria for patients with PD were as follows: PD patients who met the 2015 Clinical Diagnostic Criteria for idiopathic Parkinson’s Disease (MDS-PD Criteria) by the International Movement Disorder Society [14], disease duration ≥ 5 years, age between 18 and 75 years and indication of STN-DBS. Exclusion criteria were atypical parkinsonism, severe cognitive impairment, severe psychiatric disorders, levodopa motor response lower than 30% and contra-indications to surgery. We used the RLS diagnostic criteria of the International Restless Legs Syndrome study group (IRLSSG) in 2014 [15], which comprised the following five essential criteria: (1) a regular urge to move the legs; (2) the urge to move the legs and any accompanying unpleasant sensations that begin, or worsen, during periods of rest or inactivity; (3) the urge to move the legs and any accompanying unpleasant sensations that are partially or totally relieved by movement; (4) only occur or are worse in the evening or night than during the night; (5) the occurrences of the above features are not solely accounted for as symptoms primary to another medical or behavioral condition. We excluded secondary RLS, such as renal failure, iron deficiency anemia, myelin disease, and multiple sclerosis, etc. Exclusion criteria included not matching any of the inclusion criteria.

### 2.2. Preoperative Evaluation and Postoperative Follow-Up

Following the conventional DBS protocol, the collected PD patient information included gender, age, disease duration, Hoehn–Yahr (H–Y) grade, MMSE, MoCA, preoperative UPDRS-Ⅲ (med-on and med-off), levodopa-challenge test results, levodopa equivalent daily dose, and other clinically relevant assessments. Patients in the RLS group completed the Clinical Global Impression (CGI), Medical Outcomes Study 12-item Sleep Scale (MOS sleep), RLS Quality of Life Questionnaire (RLS QLQ), International Restless Legs Syndrome Severity Scale (IRLS), and other questionnaires before STN-DBS surgery and at 1 year after surgery. Additionally, we recorded the DBS parameters that induced RLS aggravation or relieved RLS symptoms during the follow-up process. Additionally, we recorded postoperative medication doses and postoperative UPDRS-Ⅲ. The IRLS improvement rate = (preoperative IRLS score—postoperative IRLS score)/preoperative IRLS score × 100%. We transformed the total RLS QoL score (sum over items 1–5, 7–10, and 13) to a 100% interval, with 100 representing the optimal status [16]. Data were captured by more than 3 neurosurgeons specializing in DBS evaluation of our team to guarantee their reliability and validity.

### 2.3. Neuroimaging Data

All the patients underwent cranial 3.0 T Magnetic Resonance Imaging (Siemens MAGNETOM Skyra, Germany) before surgery. T1, T2, and QSM image scan parameters are presented in the Appendix A. On the day of the surgery, after installing the Leksell head frame, we performed a preoperative 1 mm/layer head CT scan. During the operation, the patient was in a head frame, which allowed us to scan their head while they were under local anesthesia, and the scanning parameters were the same as those of the preoperative CT. Postoperative CT: within 4 days after surgery, we examined the 1 mm/layer head CT scan to exclude intracranial hemorrhage and pneumocephalus, and we combined the results with the preoperative MRI to confirm electrode contacts and contact positions. If the patient’s condition changed after surgery, such as cerebral hemorrhage, severe trauma, worsening of sudden symptoms, and failure of procedural control, further examination by cranial CT and 1.5 T MRI scan was required.

### 2.4. DBS Implant

We used the Leksell G head frame and Surgiplan system (Elekta AB, Stockholm, Sweden) to implant the electrodes. We will briefly describe the surgical procedure. The head frame was installed under local anesthesia in the ward, the bilateral STN targets were located under MRI, and their three-dimensional coordinates were calculated. In the operating room, general anesthesia, endotracheal intubation, indwelling catheterization, and disinfection of drapes were performed. According to the surgical planning system, the scalp was cut open in an arc, the skull was dried, and the subarachnoid space-sealing technique was used; this is where the dura mater is cut to prevent the loss of cerebrospinal fluid and pneumocephalus, reducing brain shift [17]. After implanting the electrodes under general anesthesia, we sealed the bone holes with biomedical fibrin glue. The electrodes were 3389 (Medtronic, Villalba, PR, USA) or L301 (PINS, Beijing, China). After fixing the electrode with Stimloc (Medtronic, Villalba, PR, USA) or Leadloc (PINS, Beijing, China), we sutured the scalp. During the operation, we did not awaken the patient to perform the macrostimulation test. Instead, we confirmed that the electrode position was accurate by merging the intraoperative CT scan with the preoperative Magnetic Resonance in the Surgiplan system. After that, we implanted an extension lead and implantable pulse generator while the patient was under general anesthesia.

### 2.5. Position of Electrodes and Contacts

Our team previously reported the electrodes and contacts reconstruction method [18]. By using Lead-DBS software (http://www.lead-dbs.org/, accessed on 6 September 2022), input preoperative Magnetic Resonance T2 and T1, and the postoperative CT scans, we linearly visualized the postoperative images and the preoperative images using the Statistical Parametrical Mapping software version 12 (SPM12) and BRAINSFit software. Then, we nonlinearly warped the images into standard stereotactic (MNI; ICBM152 2009b nonlinear asymmetric) space using a fast diffeomorphic image registration algorithm (DARTEL) [19]. We automatically prelocalized electrode trajectories, and we manually refined the results in an MNI space using Lead-DBS. This procedure allowed us to visualize the recording sites of all the patients together in one figure. After electrode and contact reconstruction, we entered patient stimulation parameters to generate the volume of tissue activated (VTA).

### 2.6. Statistical Analysis

We performed statistical analyses with SPSS26.0 statistical software (IBM SPSS Inc., Chicago, IL, USA). Measurement data that conformed to a normal distribution were expressed as mean ± standard deviation, and we analyzed these data by performing a Student’s *t*-test. Measurement data that did not conform to a normal distribution were expressed as the median (P50), and we analyzed these data using Wilcoxon rank sum tests. Regarding qualitative data, we performed a chi-square test or Fisher (If α = 0.05, and *p* < 0.05 (two-tailed)), we considered the difference between the groups to be statistically significant).

## 3. Results

### 3.1. Clinical Data Related to PD Patients STN-DBS

We enrolled 363 PD patients who met the inclusion criteria (178 men, 185 women). In total, 33 RLS patients (14 men, 19 women) and 330 non-RLS patients (164 men, 166 women) participated. Baseline data including age, disease duration, preoperative UPDRS-Ⅲ (med-on andmed-off), neuropsychiatric status examination (MMSE) results, Montreal Cognitive Assessment (MoCA) results, Hoehn–Yahr grade, and levodopa equivalent daily dose showed no differences between the RLS group and the non-RLS group. We also did not find a difference between groups based on post-operation UPDRS-Ⅲ med-on and med-off states (*p* > 0.05) (Table 1).

A total of 16 patients had complications directly related to the DBS surgery (2 in the RLS group and 14 in the non-RLS group), and they had about 1–3 mL intracerebral hemorrhage around the frontal electrode. However, none of the patients had residual permanent neurological symptoms. We observed postoperative complications in 19 patients (2 in the RLS group and 17 in the non-RLS group): 1. One non-RLS patient had postoperative delirium (persecutory delusions) for up to 1 month, and the CT scan showed no obvious intracranial hemorrhage. After treatment with olanzapine, he gradually recovered. Unfortunately, at the 1-year follow-up, his cognitive function had decreased by four points (MMSE); 2. A total of six non-RLS patients developed acute RLS symptoms the night after the STN-DBS operation (IPG-off), and additional levodopa and dopamine receptor agonists were ineffective (the patients had already taken medication before surgery). The newly emerged RLS symptoms were reduced after the administration of sedatives. The patients’ RLS symptoms spontaneously disappeared the next day, and we found no intracranial hemorrhage by CT scan; 3. One patient in the non-RLS group experienced delayed healing of the left forehead incision, which healed following our incision management procedure [20]; 4. Two patients in the non-RLS group developed a postoperative pulmonary infection and recovered with antibiotics. No patient developed an incision infection or skin erosion during the follow-up period.

### 3.2. Efficacy of STN-DBS on RLS Symptoms

#### 3.2.1. CGI Score for STN-DBS Alleviated RLS at One-Year Follow-Up

According to the Clinical Global Impression (CGI) scale results of thirty-three RLS patients, nine patients (30.30%) experienced considerable improvement (the complete or almost complete relief of all symptoms), nineteen patients (56.58%) experienced a moderate improvement (a partial reduction in symptoms), four patients (9.09%) experienced minimal improvement (a slight reduction in symptoms without changing the patient’s condition), one patient was unchanged (3.03%), and zero patients were worse. The efficacy index of STN-DBS for the treatment of RLS was 2.98 ± 0.81 (efficacy index = efficacy score/side effect score).

#### 3.2.2. IRLS, MOS Sleep and RLS QoL Scores between Pre-Operation and Post-Operation 1-Rear Follow-Up

At the 1-rear, the RLS scores of the RLS group decreased from 18.76 ± 7.71 to 8.12 ± 7.08 after STN-DBS (Z = −4.940, *p* < 0.001). The patients’ sleep quality (MOS sleep) also significantly increased at the 1-year follow-up, namely, their sleep disturbance decreased (*p* < 0.001), sleep adequacy increased (*p* < 0.001), daytime somnolence decreased (*p* = 0.005) and sleep quantity increased (*p* < 0.001) (Table 2). The time taken to fall asleep was shortened, and most patients’ sleep time at night was prolonged (*p* < 0.001). Furthermore, the STN-DBS increased patients’ RLS QLQ scores from pre-operative levels of 63.79 ± 22.60 to 86.59 ± 16.59 (*p* < 0.001) which demonstrated that the patient’s quality of daily life was ameliorated as well (Table 2).

#### 3.2.3. Changes in Anti-Parkinsonism Medication in RLS Group Pre- and Post-Operation

The levodopa equivalent daily dose (LEDD) in the RLS group decreased from preoperative levels of 814.99 ± 297.61 mg to postoperative levels of 386.42 ± 235.81 mg (t = 8.211, *p* < 0.01). Among them, the number and dosage of dopamine agonists and NMDA receptor antagonists (amantadine daily doses) were significantly reduced, but the proportion of dopamine agonists in LEDD had not significantly changed (Table 3).

### 3.3. Stimulation Parameters for the RLS Group

The 33 RLS group patients used 10 102RZs (PINS, Beijing, China) and 22 Activa RCs (Medtronic, Minneapolis, MN, USA). We used a total of 66 electrodes and 80 contacts for activation. Twenty-three patients used the bilateral monopolar mode and ten used the interleaving mode. The average amplitude of each contact was 2.51 ± 0.40 V (1.3–4.1 V); the average pulse width was 83.13 ± 3 5.46 μs (30–210 μs); and the average frequency was 120.63 ± 21.17 Hz (60–160 Hz). Among the 80 activated contacts, the 18 most inferior contacts, 38 inferior contacts, 18 superior contacts, and 6 most superior contacts were present. The electrode locations in these patients are presented in Figure 1A,B. In these 33 patients, the anatomical location of the right STN at the AC-PC was approximately 2.38 ± 0.31 mm posterior to the midpoint of the anterior–posterior joint, 11.95 ± 0.52 mm laterally, and 2.58 ± 0.36 mm inferiorly. The anatomical location of the left STN at the AC-PC was approximately 2.93 ± 0.66 mm posterior to the midpoint of the anterior–posterior joint, 11.51 ± 0.24 mm laterally, and 2.15 ± 0.11 mm inferiorly. VTA contact activation areas covered by STN nuclei reduced RLS symptoms and PD motor symptoms in these patients (Figure 2A–C). We assembled the primary activation contacts at the medial central of the sensorimotor part of the STN. The VTA covered the sensorimotor and associative parts of the STN, as well as the zona incerta (ZI).

### 3.4. Changes in RLS Symptoms during Follow-Up

At the 1-year follow-up, nine patients complained that their RLS had reappeared and came to ask for programming. The average time for RLS symptom relapse to occur was 6.78 ± 2.54 months. After programming, their RLS symptoms were reduced again, whereas the PD motor symptom scores did not change considerably. The main programming methods increased the pulse width by about 10–20 μs or the voltage by about 0.2–0.4 V at the inferior contacts of the contralateral STN of the RLS limb (seven contacts are in the central and inferior part of the sensorimotor STN; two are in the central and superior part of the sensorimotor STN). The responsive electrode contact positions for the nine patients are shown in Figure 1C. The anatomical position of the right STN at the AC-PC in these 9 patients was 2.37 ± 0.30 mm posterior to the midpoint of the anterior–posterior joint, 11.75 ± 0.54 mm laterally and 2.94 ± 0.27 mm inferiorly. The anatomical location of the left STN at the AC-PC was approximately 2.94 ± 0.14 mm posterior to the midpoint of the anterior–posterior joint, 11.08 ± 1.90 mm laterally, and 2.81 ± 1.67 mm inferiorly. The VTA contact activation area that improves RLS mainly covers the central sensorimotor area medial to the STN (Figure 2D–F). During the follow-up, we found that a total of two patients who were in the non-RLS group, who had no obvious RLS symptoms before surgery, had newly emerged RLS symptoms after the anti-parkinsonism medication reduction. Subsequently, one patient took pregabalin and the other patient restored their anti-parkinsonism medication dose to eliminate their RLS symptoms.

### 3.5. Electrical-Stimulation-Induced Acute RLS Symptoms

During our initial tuning of the patients’ DBS stimulation parameters and the follow-up programming process, we recorded five patients who had no RLS complaints before surgery but who later experienced stimulation-induced acute RLS symptoms. After reducing the voltage or pulse width, their stimulation-induced RLS symptoms disappeared. After reconstructing of the responsive contacts that cause RLS symptoms, we found that all the five contacts were in the medial inferior part of the sensorimotor part of the STN or in the substantia nigra (Figure 1D). The anatomical position of the right STN at the AC-PC in these 5 patients was −1.02 ± 0.20 mm posterior to the midpoint of the anterior–posterior joint, 13.72 ± 0.99 mm laterally and −0.55 ± 0.13 mm inferiorly. The anatomical location of the left STN at the AC-PC was approximately −1.13 ± 0.78 mm posterior to the midpoint of the anterior–posterior joint, 13.66 ± 0.53 mm laterally, and −1.77 ± 1.00 mm inferiorly. The lowest VTA contact activation area overlaps with the lower edge of the STN close to the substantia nigra (Figure 2G–I).

## 4. Discussion

### 4.1. Benefit of STN-DBS in Reducing RLS Symptoms

We found that the incidence of PD patients with RLS was 9.1%, which is similar to the results of previous reports in China [21,22], but is lower than the results of previous reports outside of China [11]. This finding may be related to the sample size and ethnic differences of the population that we investigated. This study demonstrated a significant improvement in the IRLS sum score and severity after STN-DBS in 33 patients with PD and RLS (5 with mild disease and 28 with moderate to severe disease). We also observed that most PD patients had considerably reduced RLS symptoms, which was found in previous studies [8,9,10]. Two-thirds (66.6%) of the patients improved to a mild degree of RLS symptoms (<10 on IRLS score), and 30.30% of the patients had significant improvement of symptoms (IRLS improvement rate >80%). In CGI, the efficacy index (EI) could evaluate the effectiveness of the treatment and the side effects caused by the treatment. We found a high CGI scale in the RLS group through STN-DBS (EI > 1). The RLS QoL scale and MOS sleep scale also improved postoperatively for the majority of patients with RLS when the IRLS sum score was reduced and the severity of RLS was improved. Although RLS symptoms could recur without the aggravation of PD motor symptoms, individual programming was able to maintain symptom reduction for at least one year.

### 4.2. Postoperative Medication Adjustment Strategies

Scholars [12,13] have reported that the aggravation of RLS symptoms after STN-DBS is related to a reduction in levodopa or dopamine receptor agonists. We attempted to preferentially reduce levodopa and postoperatively keep dopamine agonists unchanged in patients. If the patient did not experience RLS symptom aggravation after the LEDD reduction, we gradually reduced the dopamine agonists. However, no difference existed between the postoperative and preoperative ratio of the dopamine receptor agonists in the total LEDD. Therefore, this strategy might only be effective in some patients, or it might be overshadowed by the effect of DBS.

### 4.3. The Stimulation Coordinates according to the AC-PC Coordinates

In order to reduce the heterogeneity caused by the difference of individual skull volumes, this study used the LEAD DBS software. The midpoint coordinates of the activated contact areas in patients who belong to the RLS group were calculated by MNI coordinate transformation. The VTA covers the sensorimotor and associative parts of the STN, as well as the zona incerta (ZI), to improve RLS symptoms and PD motor symptoms. The VTA mainly covers the central sensorimotor area medial to the STN, which may be the best stimulating area for improving RLS. The VTA contact activation area that is set to overlap with the lower edge of the STN, which is close to the substantia nigra, may induce RLS symptoms.

### 4.4. Programming for RLS

The central part of the STN stimulation appeared to relieve RLS symptoms in an effective manner. Fortunately, the central part of the STN [23] and the zona incerta (ZI) [24] are the most effective areas for PD motor symptoms. Therefore, we tried to activate their contact within the medial central sensorimotor part of the STN to reduce both RLS and PD motor symptoms. If the patient had obvious dyskinesia before surgery or DBS-induced dyskinesia after surgery, we tried interleaving stimulation which activated the most superior contact to directly suppress dyskinesia [25], and maintain or slightly decrease voltage at the inferior contact. If the patient felt that the RLS symptom relief was not obvious enough, we were able to appropriately increase the pulse width of the inferior contact of the STN without causing obvious complications. Although this might not have further reduced the patients’ motor symptoms, it did relieve RLS symptoms in some patients. The stimulation parameters of STN and SN are obviously different because over-stimulation of SN can cause side effects of dizziness, nausea, and dyskinesia. Therefore, we gradually increased the stimulation amplitude from 1.5 V to avoid inducing side effects caused by electrical stimulation as much as possible. It has also been reported in the literature that combined STN-SN stimulation is superior to conventional STN stimulation in improving nocturnal RLS. We believe that the stimulation parameter settings for the central part of the STN is very important to improve RLS, but if the stimulation is too strong, it may induce leg dystonia or dyskinesia, or even RLS symptoms.

Because reducing motor symptoms was not our only programming goal, this study was slightly different from our previously report [26] in which we used a programming strategy that utilized the lowest possible stimulation parameters to obtain the same motor symptom reduction. It should be noted that when the stimulation intensity in the central sensorimotor part of the STN is too intense, leg dystonia or hypotonia may be induced, which might lead to walking problems [27]. Thus, it might be necessary to schedule more follow-up visits for RLS patients. 

### 4.5. Causes of Newly Emerged RLS after STN-DBS

RLS symptoms emerging for the first time postoperatively have been reported in previous studies [12,13]. In this study, six non-RLS patients developed RLS symptoms the night after the operation (IPG-off), and the symptoms disappeared the next day, which may have been due to the lead microlesion effect. In the other five patients who had stimulation-induced RLS symptoms, their substantia nigra or its efferent fibers might have been responsible for the induced RLS symptoms, which tended to appear immediately or within 1–2 days. Lowering the stimulation voltage or replacing it with more superior contacts might avoid the stimulation-induced RLS symptoms. Two patients in the non-RLS group reported newly emerged RLS symptoms during 3–6 months of drug reduction. These results suggest that the new RLS onset after STN-DBS was due to various reasons, which means physicians need to be careful when determining the diagnosis.

### 4.6. STN-DBS Neural Network for RLS Mitigation

The results of this study and previous studies have shown that STN-DBS [8,9,10], globus pallidus internal (GPi) pallidotomy [28], and GPi-DBS [29,30] can reduce RLS symptoms in PD patients. Ondo et al. [30] reported an extremely severe case of idiopathic RLS in a patient without Parkinson’s disease that was refractory to all pharmacological treatments. However, they improved after implantation of bilateral GPi DBS. These procedures might affect similar neural networks through the globus pallidus efferent fibers [31] in those with PD. Most of the neurons arising from the subthalamic nucleus are excitatory glutaminergic neurons and project to the GPi [32]. The GPi primarily contains inhibitory GABAergic neurons that project to the thalamus. Among them, some fibers of the sensorimotor part of the GPi correspond to the sensorimotor part of the STN and project to the ventral posterolateral nucleus. The thalamus then sends excitatory outputs to the cortex [32]. The ventral posterolateral nucleus of the thalamus not only receives projections from the globus pallidus, but also serves as a transmutation station for the spinothalamic tract, which is related to sensory afferents from the trunk and limbs. Therefore, STN-DBS may suppress RLS-related paresthesia from the spinothalamic tract in the ventral posterolateral nucleus by upregulating the globus pallidus output.

In addition to regulating the GPi, STN-DBS might also play a role in regulating the thalamus by regulating the efferent fibers of the globus pallidus. The results of this study showed that STN-DBS VTA for RLS mainly covered Ansa lenticularis and fasciculus lenticularis in the efferent pathway of the globus pallidus. Although the edges of a small number of activated volume tissues could also affect the fasciculus subthalamus, the fasciculus subthalamus mainly connects the Gpe with the anterior outer part of the STN, and it is less likely to play a role. When conducting this study, we had difficulty trying to further distinguish whether Ansa lenticularis or fasciculus lenticularis played a major role in the results. VTA is the area of brain tissue regulated by electrical pulses around the electrode contacts, that is, the volume of activated brain tissue. We used the Lead-DBS image reconstruction method to input the patient’s stimulation parameters to generate the VTA. We found that STN-DBS improved the RLS coverage area mainly in the STN sensorimotor part and undefined zone, such as the VTA stimulated in 33 people (Figure 2A–C). Factors affecting the coverage and shape of the VTA on the STN mainly include electrode contact position, stimulation parameter settings, target brain tissue anatomy, and multi-contact electrodes, etc. Therefore, precise regulation of VTA may optimize the neuromodulatory effects of DBS, improve the efficacy of DBS, and reduce adverse reactions. In the future, it is of great research significance to explore the optimal stimulation area for STN-DBS to improve PD and RLS.

STN-DBS may regulate substantia nigra function and its efferent fibers. First, RLS patients often have abnormal dopamine metabolism, and supplementation with levodopa or dopamine receptor agonists can relieve RLS symptoms [33]. The substantia nigra has projections of dopaminergic neurotransmitters to the striatum and caudate nucleus [23], and the striatum and caudate nucleus are also the most common sites of stroke-induced acute RLS [34]. In this study, we found the VTA, which could reduce RLS, covered the superior part of the substantia nigra, as well as the substantia nigra efferent fibers which emanate from the superior lateral substantia nigra and below the STN [35]. However, the contacts that are responsible for inducing RLS symptoms are closer to the substantia nigra, so the abnormality of substantia nigra function and fiber projection appears to be related to RLS symptoms. Due to the electrode trajectory angle, the substantia nigra pars compacta (SNpc) was closer to the electrode contact, whereas the substantia nigra pars reticulata (SNpr) and substantia nigra efferent fibers tended to be slightly farther from the lowermost contacts. Unfortunately, we still had a difficult time distinguishing which structure played a major role in this study, and directional electrode contacts would be a useful research tool in future studies.

### 4.7. Study Limitations

This study has two main limitations that could be improved. Firstly, this study is a retrospective study. We mainly based the setting of the stimulation parameters on the reduction of motor symptoms of Parkinson’s disease. Therefore, scholars need to preferentially set up the stimulation parameters according to RLS symptoms in the future, as the effective activation domain could be located more easily. Secondly, the use of directional electrodes in the future will enable more precise distinctions of RLS-related neural networks, whether they reduce RLS or aggravate the RLS structure.

## 5. Conclusions

We found that STN-DBS can reduce restless legs syndrome in patients with Parkinson’s disease. Patients sustained this relief over a one-year period. In addition, we showed that the central sensorimotor part of the STN is the optimal activation region for RLS symptoms in PD patients.

## Figures and Tables

**Figure 1 brainsci-12-01645-f001:**
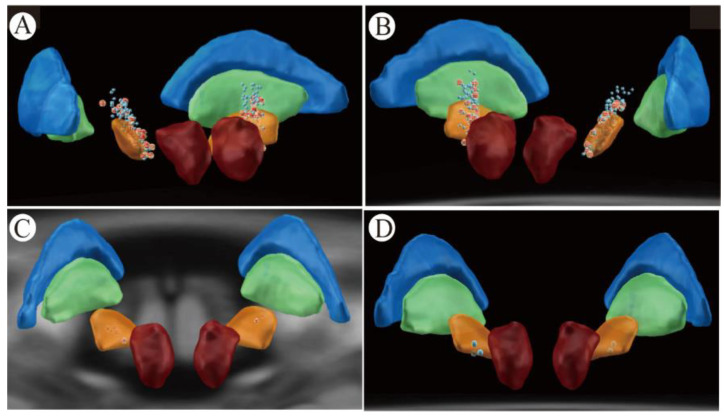
Location of effective stimulation contacts for STN-DBS to reduce RLS. (**A**) (left posterior—anterior view), and (**B**) (right posterior—anterior view) show the locations of all activated contacts (red spheres) and inactive contacts (blue) for 33 patients. (**C**) (posterior–anterior view) shows effective programmed contacts in nine patients with exacerbated RLS symptoms. (**D**) shows activation contact locations in five patients with acute RLS symptoms induced by electrical stimulation. In the figures, the yellow sphere is the subthalamic nucleus, the green sphere is the globus pallidus internal segment, the blue sphere is the globus pallidus external segment (GPe), and the red sphere is the red nucleus.

**Figure 2 brainsci-12-01645-f002:**
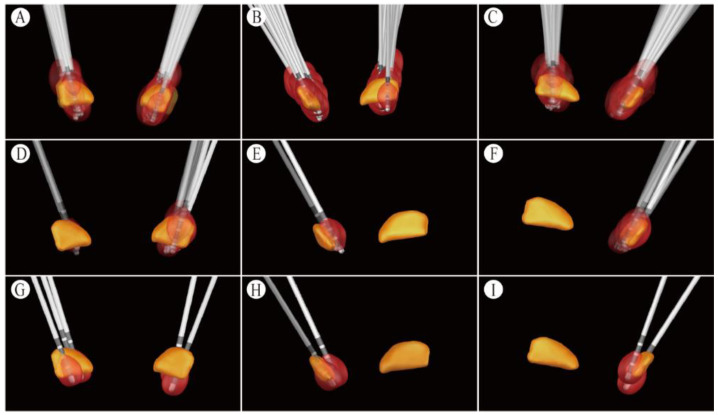
Overlapping of VTA contact activation area and the STN (including frontal, left, and right lateral views, respectively). (**A**–**C**) VTA contact activation area covered by the STN nucleus in 33 patients. The VTA covered the sensorimotor and associative parts of the STN, as well as the zona incerta (ZI). (**D**–**F**) In 9 patients with recurrent RLS, after program-controlled adjustment of stimulation parameters, RLS symptoms were significantly improved, while PD motor symptom scores did not change significantly. The VTA contact activation area at this point primarily covers the central sensorimotor area medial to the STN. (**G**–**I**) Activation of the lowermost VTA contact area induced RLS symptoms in 5 patients. The area overlapped with the lower border of the STN, and the common activation area is close to the substantia nigra.

**Table 1 brainsci-12-01645-t001:** Demographic characteristics of RLS and Non-RLS groups.

Items	RLS Group (*n* = 33)	Non-RLS Group (*n* = 330)	*p*-Value
Age	62.97 ± 6.41	62.15 ± 7.93	0.074
Disease duration	9.45 ± 4.21	10.66 ± 4.27	0.891
Gender(M/F)	14/19	164/166	0.426 a
UPDRS-Ⅲ(pre-OP, med-off)	58.09 ± 14.60	59.08 ± 17.26	0.283
UPDRS-Ⅲ(pre-OP, med-on)	24.55 ± 10.09	27.58 ± 13.76	0.161
LCT (%)	57.56% (P50), 57.25 ± 14.80	55.00 (P50), 54.69 ± 16.28	0.307
H-Y (1.5/2/2.5/3/4/5) Grade	0/1/6/18/8/0	3/11/62/191/62/1	0.968 b
LEDD pre-OP	800.00 (P50), 810.52 ± 297.61	800.00 (P50), 821.35 ± 439.02	0.976 c
UPDRS-Ⅲ (post-OP, med-off, IPG-off)	49.50 (P50), 50.57 ± 17.91	50.00 (P50), 51.08 ± 16.63	0.802
UPDRS-Ⅲ (post-OP, med-off, IPG-on)	25.00 (P50), 27.37 ± 11.39	26.00 (P50), 27.38 ± 11.67	0.930
UPDRS-Ⅲ (post-OP, med-on, IPG-off )	22.00 (P50), 21.18 ± 10.12	21.00 (P50), 21.54 ± 9.62	0.851

LCT, L-dopa challenge test; H–Y, Hoehn–Yahr grade; LEDD, L-dopa equivalent daily dose; MMSE, mini-mental state examination; MoCA, Montreal Cognitive Assessment. a: Pearson’s chi-squared test; b: Fisher’s exact test; c: Mann–Whitney U-test.

**Table 2 brainsci-12-01645-t002:** The IRLS, MOS sleep and RLS QoL scores pre- and 1 year post-operation.

Items	Pre-Operation (Mean ± sd, P50)	Post-Operation (Mean ± sd, P50)	*p*-Value
IRLS			
Discomfort	2.18 ± 0.85	1.09 ± 0.77	<0.001
Need to move	1.91 ± 0.84	0.76 ± 0.87	<0.001
Relief	1.91 ± 0.88	1.30 ± 1.16	0.002
Sleep disturbance	1.85 ± 1.00	0.70 ± 0.77	<0.001
During the day (Tiredness or sleepiness)	1.39 ± 1.06	0.33 ± 0.54	<0.001
RLS on the whole	1.85 ± 1.00	0.73 ± 1.01	<0.001
How often	2.39 ± 1.17	1.27 ± 1.26	<0.001
How severe	2.03 ± 0.88	0.97 ± 0.85	<0.001
Daily activities	1.88 ± 0.99	0.61 ± 0.79	<0.001
Mood disturbance	1.36 ± 0.99	0.36 ± 0.60	<0.001
IRLS sumscore	18 (P50), 18.76 ± 7.71	16 (P50), 8.12 ± 7.08	<0.001 *
MOS sleep			
Sleep disturbance	52.17 (P50), 57.44 ± 18.28	34.78 (P50), 40.45 ± 15.73	<0.001
Sleep adequacy	50.00 (P50), 49.75 ± 14.51	66.67 (P50), 69.95 ± 13.49	<0.001
Daytime somnolence	83.33 (P50), 80.30 ± 15.90	88.89 (P50), 85.52 ± 10.29	0.005
Snoring	16.67 (P50), 26.04 ± 20.27	16.67 (P50), 21.72 ± 14.72	0.072
Shortness of breath or headache	16.67 (P50), 25.76 ± 16.71	16.67 (P50), 22.73 ± 13.70	0.109
Sleep quantity	5.00 (P50), 4.82 ± 1.16	6.00 (P50), 5.94 ± 1.32	<0.001
RLS Quality of Life Questionnaire			
RLS QoL transformed score (1–5, 7–10, 13 items)	70.00 (P50),63.79 ± 22.60	92.50(P50),86.59 ± 16.59	<0.001 †

*: the difference between the preoperative IRS total score and the postoperative IRLS total score did not conform to a normal distribution; Wilcoxon signed-rank test for two correlated samples, *p* < 0.01. †: RLS quality of life transformed scores = [(actual raw score − lowest possible raw score)/possible raw score range] × 100. Higher score = increased quality of life.

**Table 3 brainsci-12-01645-t003:** The list of drugs used in pre- and postoperative treatment for the RLS group.

Items	Pre-Operation (P50, Mean ± sd)	Post-Operation (P50, Mean ± sd)	*p*-Value
Levodopa equivalent daily dose (LEDD)	814.99 ± 297.61	386.42 ± 235.81	<0.001 #
Total Levodopa and COMT dose	600 (P50), 665.75 ± 264.96	300 (P50), 323.54 ± 170.8	<0.001
Dopamine agonist (DA)	75.00 (P50), 80.30 ± 65.10	25 (P50), 38.64 ± 45.11	0.001
Amantadine daily doses	0 (P50), 59.09 ± 97.99	0 (P50), 21.21 ± 69.63	0.017
Total MAO-B dose	0 (P50), 9.85 ± 27.91	0 (P50), 3.03 ± 17.41	0.109

#: *t*-test. We calculated the other items with the Wilcoxon rank-sum test.

## Data Availability

The authors declare that all data supporting the findings of this study are available within the manuscript or are available from the corresponding authors upon request.

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
