# Peer review of "Optimal Contact Position of Subthalamic Nucleus Deep Brain Stimulation for Reducing Restless Legs Syndrome in Parkinson’s Disease Patients: One-Year Follow-Up with 33 Patients"

_brainsci, 2022, doi:10.3390/brainsci12121645_

Round 1

Reviewer 1 Report

Thank you for submission the great article.

A few things seem to need to be corrected.

30-32:

First, it seems necessary to mention most common neurodegenerative disease.

41:

It seems necessary to change it to RLS in the abbreviation.

59-60:

I think it would be better to move toward the "method."

Author Response

Response to Reviewer1 Comments

Point 1: 30-32:

First, it seems necessary to mention most common neurodegenerative disease.

Response 1: Thanks to the reviewers' suggestions. We have deleted second,and replaced disease with bradykinetic disorder. (34-35)

Point 2: 41:

It seems necessary to change it to RLS in the abbreviation.

Response 2: We have changed it to RLS in the abbreviation. (51)

Point 3: 59-60:

I think it would be better to move toward the "method."

Response 3:
“This study was approved by the First Affiliated Hospital of Naval Medical University Ethics Committee (CHEC2018-022)” has been moved to lines 63-64 of the methodology section.(77-78)

Reviewer 2 Report

The authors report on the optimal electrode position within the STN in case of restless legs syndrome in 33 patients.

This is an intersting topic as the definition of a "sweat spot" in the STN is of interest, even though usually not focussing on RLS, it is intersting to evaluate this condition too as diverse symptom spectra might require different localizations.

However, there are several major concerns about the manuscript.

- The authors use 1 year follow up and suggest the electrode position doenst change over time. How is this supported? Please introduce electrode stability within the introduction section. 

- The methods section needs revision. Please cleary state incluce and exclude criteria and how study groups are defines (RLS vs nonRLS?),  even though only the RLS patients are further investigated. Is the only CT after surgery always 4 days post operative or are there any other follow up scans? The surgical procedure needs to be described further. Were contacts only localized in RLS patients? this is not clear.

The authors use electrode contact position of 4 day post op scan and evaluated patients 1 year after surgery. Nine patients underwent adaption of stimulation settings in between. Were there severe changes or loss of effect suggesting electrode shift or something similar? Please justify your approach and the limitations of this. 

The results should provide some clinical cases underpinning the overall results. In addition the contact position is one aspect, but the clinical effect is also dependent on stimulations settings and the resulting VTA. What about VTA coverage of the STN? which area is typically covered?

Author Response

Response to Reviewer2 Comments

Point 1: The authors use 1 year follow up and suggest the electrode position doenst change over time. How is this supported? Please introduce electrode stability within the introduction section. 

Response 1: Thanks to the reviewers' suggestions. The intracranial electrodes are fixed with the new electrode fixation device after implantation into the bilateral SNT nucleus target, which is more stable than the traditional electrode fixation device. It is less likely that the intracranial electrodes would be displaced in the brain tissue after surgery unless they were severely traumatized or improperly operated on by the surgeon, etc. The following points can support that the electrode position will not change over time: 1) we ensured that there was no obvious pneumocephalus or  intracerebral hematoma compression displacement during the re-examination of CT; 2) 3D reconstruction of electrodes and contacts clearly shows that the electrode position is good; 3) after 1 month on, the patient's symptoms do not change such as sudden aggravation or large fluctuation of symptoms at 1-year follow-up; 4) exclude serious trauma and other cranial surgery histories, etc. Previous literature reported that there is minimal electrode displacement, and then our two previous articles analyzing the electrode position also illustrated the unlikelihood of electrode displacement in the brain tissue of patients during the follow-up period.[1]

  1. Yang C, Qiu Y, Wu X, Wang J, Wu Y, Hu X. Analysis of Contact Position for Subthalamic Nucleus Deep Brain Stimulation-Induced Hyperhidrosis. Parkinsons Dis. 2019; 2019:8180123. doi: 10.1155/2019/8180123.

Point 2: The methods section needs revision. Please cleary state incluce and exclude criteria and how study groups are defines (RLS vs nonRLS?), even though only the RLS patients are further investigated. Is the only CT after surgery always 4 days post operative or are there any other follow up scans? The surgical procedure needs to be described further. Were contacts only localized in RLS patients? this is not clear.

Response 2: The inclusion and exclusion criteria were clarified and the grouping method (RLS vs. non-RLS) for this study was detailed (79-102 rows). The main purpose of postoperative cranial CT review in all patients was to clarify whether the electrode implantation was well positioned and to exclude intracranial hemorrhage, which was usually done within 4 days postoperatively at our center. We used the subarachnoid closure technique, and bioprotein glue to close the bone holes during the procedure to avoid cerebrospinal fluid loss and pneumocephalus. We have added a description of the detailed surgical procedure. In addition to the adjustment of electrode contacts, electrical pulse parameters, electrode position, and target brain tissue factors can affect the outcome of RLS patients during BDS. In this study, we focused on the improvement of RLS symptoms by electrode contacts and voltage stimulation parameters, so we reconstructed only the electrodes and contacts of patients in the RLS group. We added this section for explanatory notes. (137-144 rows)

Point 3:
The authors use electrode contact position of 4 days post op scan and evaluated patients 1 year after surgery. Nine patients underwent adaption of stimulation settings in between. Were there severe changes or loss of effect suggesting electrode shift or something similar? Please justify your approach and the limitations of this. 

Response 3: The study Used Lead-DBS software, preoperative MRI T2, T1, and postoperative CT for fusion and image reconstruction to clarify the location of the patient's implanted electrodes, and then the range of stimulation areas activated by the contacts is determined based on the patient's programmed parameter settings. During the 1-year follow-up period, if the patient has no trauma or severe symptom fluctuations, the electrode position will not change significantly, and the CT or 1.5 TMRI will be reviewed as necessary after the procedure depending on the patient's condition. Therefore, the patient's electrode position remains unchanged and the absence of a CT review usually does not affect the patient's outcome evaluation. At the 1-year follow-up, 9 patients had recurrent RLS but DBS still controlled the patients' core PD symptoms well, so there was no reason to think that the electrodes had been shifted. We did not review cranial CT and fuse it with preoperative MRI again in these 9 patients mainly because 1. the procedure itself did not cause significant pneumocranial shift; 2. these 9 patients had no history of head trauma, no symptoms of chronic subdural hematoma or changes in PD symptoms with electrode shift; 3. the exacerbation of RLS occurred at 6.78 ± 2.54 months, and the possibility of electrode shift is low. We did not review the CT in order to not waste patient health insurance costs and to not reduce patient satisfaction. However, we do not think that this would reduce the credibility of the study results. Titration adjustment of stimulation parameters. We tried contact replacement, frequency change, increased voltage, and pulse width after efforts in order to keep the control of core PD symptoms by DBS as unchanged as possible while not increasing the isokinetic movement and improving RLS symptoms. (For the recurrence of RLS symptoms, our main program control method was to increase the pulse width or increase the voltage of the lower STN contact on the opposite side of the RLS limb.) The literature reports that STN+SN improves restless legs, suggesting that proximity to the substantia nigra may also improve RLS. The rate of improvement of RLS was significantly increased after several sessions of programmed control and selection of appropriate contacts and stimulation parameters. The limitation of this method of adjusting electrical stimulation parameters is that it requires a lot of time to try different combinations of stimulation parameters. (128-134 rows).

Point 4: The results should provide some clinical cases underpinning the overall results. In addition,the contact position is one aspect, but the clinical effect is also dependent on stimulations settings and the resulting VTA. What about VTA coverage of the STN? which area is typically covered?

Response 4: The clinical efficacy of DBS depends not only on the contact location but also on the stimulation parameter settings and the generated VTA, which is the area of brain tissue around the electrode contact that is modulated by electrical pulses, i.e., the volume of activated brain tissue. Through the Lead-DBS imaging reconstruction method, the patient's stimulation parameters are input to generate activated volumetric tissue (VTA) after electrode and contact reconstruction. Precise adjustment of VTA may optimize the neuromodulatory effects of DBS, enhance the efficacy of DBS, and reduce adverse effects. The main factors affecting the coverage and shape of VTA on STN include electrode contact position, electrical pulse parameter settings (stimulation parameters), brain tissue anatomy in the target area, multi-contact electrodes, etc. The area usually covered by the therapeutic effect of STN-DBS consists of 3 main parts: (1) anterodorsal region of STN (2) undefined zone (3) white matter bundle. The accuracy of the electrode position is key to ensure the efficacy of DBS, and is also a prerequisite for a good postoperative program control parameter setting. The key to program control is to set the optimal contacts according to the various symptoms of the patient. Therefore, it is of great research significance to explore the optimal stimulation area for STN-DBS to improve PD and RLS. In this paper, we add VTA maps of patients with improved RLS, as well as the VTA maps of patients with induced RLS. (356-364 rows).

Finally, we apologize for the poor language of our manuscript. We worked on the manuscript for a long time and repeatedly revised sentences. We have now worked on both language and readability and have also involved native English speakers for language corrections.

Reviewer 3 Report

The authors present their Experiences  regarding the effects of STN-DBS in patients with PD on RLS. Of  363  patients undergoing STN-DBS 33  suffered from RLS prior to surgery. The efficacy on RLS symptoms  is assessed according  to the CGI score and IRLS. Adverse events, stimulation parameters and contact localization were reported.

We have to congratulate the authors to this concise manuscript on a relevant topic. We would recommend some minor changes:

- please mention the MCID of CGI and IRLS and discuss this further

- Please mention the stimulation coordinates according to the AC-PC coordinates. 

- Please discuss the possibility of different stimulation parameters in STN und SN to address both RLS and PD motor symptoms.

- Is there room for DBS in RLS without PD?

We would recommend  the  manuscript for publication after minor revision.

Author Response

Response to Reviewer3 Comments

Point 1: please mention the MCID of CGI and IRLS and discuss this further.

Response 1: IRLS, as the primary observation for evaluating RLS, and the efficacy index (EI) in CGI can evaluate the effectiveness of treatment and treatment-induced side effects, which are added in the discussion section 4.1.(331-342 rows)

Point 2: Please mention the stimulation coordinates according to the AC-PC coordinates. 

Response 2: In 33 patients, the anatomical location of the right STN at the AC-PC was approximately 2.38±0.31 mm posterior to the midpoint of the anterior-posterior joint, 11.95±0.52 mm laterally, and 2.58±0.36 mm inferiorly. The anatomical location of the left STN at the AC-PC was approximately 2.93±0.66 mm posterior to the midpoint of the anterior-posterior joint, 11.51±0.24 mm laterally, and 2.15±0.11 mm inferiorly. The anatomical position of the right STN at the AC-PC in 9 patients was 2.37±0.30 mm posterior to the midpoint of the anterior-posterior joint, 11.75±0.54 mm laterally and 2.94±0.27 mm inferiorly. The anatomical location of the left STN at the AC-PC was approximately 2.94±0.14 mm posterior to the midpoint of the anterior-posterior joint, 11.08±1.90mm laterally, and 2.81±1.67 mm inferiorly. In summary, the coordinate range of the anatomical position of the STN in the AC-PC are as follows: the midpoint of the anterior-posterior joint is about-0.10-5.79 mm posteriorly, 10.03-14.92 mm laterally, and -2.52-6.12 mm inferiorly. This section was added to the discussion section. (251-257 rows, 271-277 rows, and 291-296 rows, and 356-365 rows).

Point 3: Please discuss the possibility of different stimulation parameters in STN und SN to address both RLS and PD motor symptoms.

Response 3: The stimulation parameters of STN and SN are obviously different because over stimulation of SN can cause side effects of dizziness, nausea, and dyskinesia. the stimulation voltage of STN is 2.51±0.07 (n=33), and there are patients using crossed electrical pulses, where the pulse width and frequency are not easy to count. Therefore, we gradually increased the stimulation amplitude from 1.5V to avoid inducing side effects caused by electrical stimulation as much as possible. It has also been reported in the literature that combined STN+SNr stimulation is superior to conventional STN stimulation in improving nocturnal RLS. (379-387 rows)

[1] Hidding U, Gulberti A, Pflug C, Choe C, Horn A, Prilop L, et al. Modulation of specific components of sleep disturbances by simultaneous subthalamic and nigral stimulation in Parkinson's disease. Parkinsonism Relat Disord. 2019;62:141-7. doi: 10.1016/j.parkreldis.2018.12.026.

Point 4: Is there room for DBS in RLS without PD?

Response 4: Yes, DBS is an effective method capable of relieving clinical symptoms in patients with refractory RLS, including primary RLS. DBS has also shown significant efficacy in restless legs syndrome in the absence of Parkinson's disease. For example, Ondo et al [2] reported an extremely severe case of idiopathic RLS in a patient without Parkinson's disease that was refractory to all pharmacological treatments. However, they improved after implantation of bilateral GPi DBS. Reference 27 of this article only mentions that GPi DBS improves RLS, but does not discuss in detail the therapeutic effect of DBS on idiopathic RLS. The revised manuscript adds discussion in this regard. Although current studies only reported that GPi DBS improves idiopathic RLS, STN-DBS has not been reported. However, DBS is well accepted by clinicians and patients, and the therapeutic efficacy and effective targets of DBS for primary RLS could be explored in the future through prospective studies or multiple research centers jointly. (413-416 rows)

[2]   Ondo W G, Jankovic J, Simpson R, et al. Globus pallidus deep brain stimulation for refractory idiopathic restless legs syndrome[J]. Sleep Med, 2012,13(9):1202-1204.

Round 2

Reviewer 2 Report

The revised version covered all mentioned aspects of the review.